# Microfluidic Immobilized Enzymatic Reactors for Proteomic Analyses—Recent Developments and Trends (2017–2021)

**DOI:** 10.3390/mi13020311

**Published:** 2022-02-17

**Authors:** Cynthia Nagy, Ruben Szabo, Attila Gaspar

**Affiliations:** Department of Inorganic and Analytical Chemistry, University of Debrecen, Egyetem ter 1., 4032 Debrecen, Hungary; nagy.cynthia@science.unideb.hu (C.N.); szabodavidruben@gmail.com (R.S.)

**Keywords:** 2017–2021, microfluidic, enzyme reactor, particle, monolith, enzyme immobilization, protein digestion

## Abstract

Given the strong interdisciplinary nature of microfluidic immobilized enzyme reactor (μ-IMER) technology, several branches of science contribute to its successful implementation. A combination of physical, chemical knowledge and engineering skills is often required. The development and application of μ-IMERs in the proteomic community are experiencing increasing importance due to their attractive features of enzyme reusability, shorter digestion times, the ability to handle minute volumes of sample and the prospect of on-line integration into analytical workflows. The aim of this review is to give an account of the current (2017–2021) trends regarding the preparation of microdevices, immobilization strategies, and IMER configurations. The different aspects of microfabrication (designs, fabrication technologies and detectors) and enzyme immobilization (empty and packed channels, and monolithic supports) are surveyed focusing on μ-IMERs developed for proteomic analysis. Based on the advantages and limitations of the published approaches and the different applications, a probable perspective is given.

## 1. Introduction

Microreactors are small devices consisting of micrometer-wide capillaries or channels. Such devices are designed to carry out a range of biological and chemical reactions with the inherent advantages of less reagent consumption, flexible and well-controllable operation and simple integration with other units. A common feature of microreactors is their high specific surface area, which can enable a fast reaction rate.

The utilization of enzymes in reactors has been increasing in the last few decades, especially immobilized enzyme reactor (IMER) applications, where the enzymes are confined to a solid support [1,2]. Although these reactors can be assembled from conventional laboratory devices, such as tubes, valves or reactor chambers, these reactors can also be miniaturized and transferred to a microchip format. In such microfluidic or microchip IMERs (μ-IMERs) [3,4] not more than a few microliters of sample or reagent are used and those are not larger than a few tens of cm^2^. The immobilization of enzymes offers the possibility of reusability, simple handling, easy separation of products from the enzyme and increased stability of the enzymes to changes in operational conditions.

The developments and applications of μ-IMERs have been receiving a tremendous amount of attention due to their advantages over the traditional, larger analytical systems. These advantages include advanced heat and mass transfer, high surface-to-volume ratio (S/V), enhanced catalytic efficiency, reduced diffusion distance, and high operational safety [5]. The operational costs in μ-IMERs are typically quite low, as the consumption of the enzymes can be strongly decreased by immobilization. Since these microfluidic devices are often cheap and disposable, their maintenance or regeneration can be avoided [6]. The high enzyme-to-substrate ratios achievable in the μ-IMERs improve the digestion efficiency even for low-abundance proteins. Because the enzymatic reaction is carried out under liquid flow, the reagents and the products are continuously removed from the surface of the reactor, thus, the catalytic process is not inhibited. A further benefit of the use of IMERs over the application of entire living cells is that easier purification processes are required (fewer by-products or contaminants, or no cellular debris are obtained) [7]. In chips, according to the original initiative of the lab-on-a-chip conception, several consecutive steps might be integrated, which can either be reactors with different immobilized enzymes or the IMER is integrated into other microfluidic units (for separation, enrichment, derivatization, detection, etc.) [8,9]. The advantages of IMERs regarding the short reaction/analysis time and high efficiency in catalytic reactions were thoroughly discussed in many papers and reviews [3,4,10,11]. Perhaps the largest drawback of microfluidic reactors is the limited amount of components produced in the device, which can be mitigated by parallelization of channel/reactor systems. On the other hand, in several fields (e.g., chemical informatics, identification, analysis) the submicrogram amount of components is still tolerable.

Microfluidic chips satisfy the most important requirement for high IMER efficiency, which is the large specific surface area (S/V ratio) of solid supports. From this point of view there are three main types of enzyme reactors: (1) wall-coated IMERs, where the enzyme is directly adsorbed/attached on the inner surface of the empty [12,13] or micropatterned (e.g., micropillar array [14]) microchannels; (2) packed/fixed-bed IMERs, where the enzyme is immobilized to a support material (particles, beads) that can be homogeneously packed into the microfluidic system [15,16]; and (3) monolithic IMERs, where the enzyme is immobilized onto the microscopic pores and channels provided by the network of the meso- and macro-porosity of a monolith-type material [17,18] (Figure 1). There are several other solid supports used in μ-IMERs, which apply slightly other approaches through membrane- [19,20], paper- [21] or gel-based [22,23] supports. Although the simplest realization of enzyme immobilization can be achieved on the channel interior/wall itself, the most often used classical way to increase the S/V of the support is the application of a micropacking or membrane. Such a great variety of microreactor designs implies variability in performance, as well. For the assessment and comparability of microreactors, a set of parameters (residence time; enzyme load; (specific) enzyme activity; substrate concentration; reactor size, productivity and stability) should be specified [24], however, not all publications report these key parameters.

The μ-IMERs are used almost exclusively for analytical aims. The majority of the applications are proteomic or glycomic related studies [23,25,26,27], where the immobilized proteolytic enzyme is used for the digestion of the investigated proteins. In some works, the immobilized enzyme assists in the transformation of analytes to components that can be more efficiently or sensitively detected [28,29,30,31,32,33,34].

In the last 5 years, the number of articles annually published about μ-IMERs in scientific journals was close to one hundred (Figure 2) and more than 10 reviews summarizing these works appeared in the field (Table 1). In this review, the different aspects of microfabrication (designs, fabrication technologies and detectors) and enzyme immobilization (empty and packed channels, and monolithic supports) are surveyed focusing on μ-IMERs developed for proteomic analysis. Based on the advantages and limitations of the published approaches and the different applications, a probable perspective was tried to be concluded.

## 2. Fabrication of IMERs

The proper reactor design and configuration should be selected depending on (1) which platform (capillary, microfluidic channel/device), (2) on what support (packed microbed, wall-coated or monolithic reactor), and (3) how (adsorption, chemical bondings or bioaffinity interactions) the enzymes are immobilized.

### 2.1. Designs, Materials and Fabrication Technologies

The capillary- and chip-based devices are the main types of microfluidic reactors. Capillary-based reactors can be easily scaled up by lengthening the capillary, and those can be easily connected with other microfluidic devices or standard separation and detection techniques (chromatography, electrophoresis or mass spectrometry). The chip reactors provide more complexity and flexibility in the design of channel patterns and no dead volume between the parts of the fluid systems should be expected. While chips are truly smaller than a few cm^2^, the capillary-based systems are often longer in one dimension. Both capillaries and chip channels can be parallelized.

A frequent problem arising when handling microfluidic devices is the clogging of channels or microscopic patterns, which can be resolved by channel designs, including by-pass routes if a part of the channel is blocked, the application of a high purging pressure (if the material of the device is durable), or by simply discarding the device if its low price allows disposability.

The material of the μ-IMER must be compatible with the methodologies of enzyme immobilization and the efficient working conditions for the enzymatic function. Most of the capillary-based μ-IMERs utilize commercially available fused silica capillaries to accommodate the enzymes. A large variety of methods exist for the derivatization of these capillaries, which mostly rely on surface-modification [46,47,48,49] or in situ monolith formation [50,51,52,53,54,55,56,57]. The greater flexibility regarding reactor layout and architecture in the case of chips is enabled by (often) in-house created devices made by means of microfabrication. Initially, hard materials, such as silicon, quartz or glass were used for manufacturing chips, where the channel systems were patterned by etching techniques. Glass microchips are still being used [50,58,59], however, softer, polymer-based materials, e.g., polydimethylsiloxane (PDMS) [13,14,60] have also attracted considerable attention. PDMS is one of the most broadly used polymers due to its low cost, biocompatibility, optical transparency and flexibility. However, its use is limited to the academic research culture, presumably because of its low mechanical durability. Furthermore, PDMS is highly hydrophobic and lacks surface functional moieties, therefore, surface treatment is necessary to prevent the non-specific adsorption of molecules. On the other hand, this supreme absorptivity can actually be exploited for the direct immobilization of enzymes [13,61]. Besides PDMS, thiol-ene (TE) microchips are experiencing increasing interest [60,62,63,64]. In addition to optical transparency and low price, TE chips offer improved solvent resistance and the possibility of tuning the surface chemistry by modulating the stoichiometry of the monomers. The fabrication of such PDMS and TE-based chips is a straightforward procedure based on the principle of replica molding. Microchips are created by making replicas of a master mold, which contains the desired channel structure. The master mold is prepared in advance by using, e.g., photolithography [13,62] or high precision milling [64]. Polymer-based microchips can also be fabricated by the direct patterning of the material, as realized by Wouters and coworkers, who created cyclic-olefin-copolymer (COC) microchips using micromilling [65].

Although the topic of this review is microfluidic IMERs, it is worth highlighting the importance of nanofluidic chips, since such devices can have great contributions to the emerging field of single-cell shotgun proteomics. Yamamoto et al., devised a glass-based chip, in which the nanochannels were manufactured by electron-beam lithography and etching [59]. E-beam lithography allows higher resolution than photolithography, since the resolution is not restricted by the diffraction of photons, however more sophisticated instrumentation is required.

### 2.2. Coupling to Downstream Processing Units

In general, the μ-IMERs used for proteomics should be hyphenated with separation methods and MS (mainly ESI-MS). The majority of the published works describe microreactors used off-line before the subsequent proteomic analysis. Although the on-line coupling of μ-IMERs is still challenging, as it often requires complex instrumental setups, and numerous research managed the on-line coupling with LC- or CE-MS [48,64,66]. The accomplishment of an on-line μ-IMER-LC-MS or CE-MS system provides minimal dead volume and contamination, the processing of submicroliter volumes of the sample with high efficiency, and high throughput and automated analysis.

The biggest difficulty in developing a reliable on-line coupled μ-IMER-separation-MS-detection system is the harmonization of appropriate experimental conditions (solvent, pressure or electrical field) of each respective unit in the workflow. In the case of microchip IMERs, limitations are imposed regarding the choice of chip material. PDMS cannot withstand pressures greater than ~2 bars, however, Jönsson and coworkers demonstrated the possibility of operating TE chips up to ~34 bars, thus presenting an elegant on-line μ-IMER-LC-MS platform. A 3D printed chip holder was used to interface the chip with the LC-MS system [64]. Since LC-MS and CE-MS apparatus are commercially available on-line coupled systems, the main difficulty lies in creating an automated, continuous fluidic connection between the μ-IMER and the separation platform. The on-line coupling of digestion and LC separation can be achieved with the use of switching valves [46,47,56,66,67,68]. In such configurations, the μ-IMER and the analytical column are connected via a valve, where the microreactor can be considered as the first dimension of the setup. In another approach, Wilson et al., showed the seamless integration of capillary-based microreactors, where the μ-IMER was attached to a 6-port valve as a loop [56,66].

While creating the μ-IMER in the separation unit itself does not seem a viable approach in the case of LC, mostly because of the inconsistency in the pressure requirement of each process, CE can provide the means of satisfying such endeavors, forming an in-line strategy. It is possible to form the reactor part only on a short, initial section of the separation capillary [48,52,57,69]. Once the sample protein is injected, the sample is either parked or transported through the reactor section and upon application of voltage the products are separated in the remaining part of the capillary. Such workflows obviously necessitate the use of electrolyte systems compatible with both the enzymatic reaction and the CE separation. Furthermore, in cases where ESI-MS detection is used, solvent volatility is also an issue; the application of volatile background electrolytes is strongly advised for the proper transfer of analytes into the gas phase. An overview of the most common on-line configurations mentioned above is shown in Figure 3.

The majority of works dealing with proteomic μ-IMERs utilize MS(/MS) detection due to its ability to provide valuable information about peptide identity and possible post-translational modifications (PTMs). In the field of proteomics, two types of ion sources can be used: electrospray (ESI), and matrix-assisted laser desorption/ionization (MALDI) source. Since MALDI requires the analytes to be deposited on a plate and co-crystallize with a special matrix prior to analysis, it is not possible to carry out on-line hyphenation with upstream separation workflows, since those operate in continuous-flow mode. Nevertheless, a number of works have demonstrated the off-line use of MS equipped with a MALDI ion source [49,55,68]. The utilization of ESI (and especially nanoESI) interfaces prevail in the literature, one of the main reasons being their compatibility with upstream flow systems, which enables on-line coupling. Several on-line integrated μ-IMER platforms have been published for both LC [47,64,66,67,68] and CE-related works [48,69], as well as off-line systems [13,14,51,54,55,58,60,65,70] with ESI-MS detection.

Apart from MS, UV detection is also popular [13,48,49,52,53,55,56]. UV detection is exceptionally practical in cases where CE separations are conducted since its configuration enables on-capillary detection. Therefore, a truly in-line workflow can be developed, where proteolysis, separation and detection are all carried out in a single capillary [52].

## 3. Immobilization of Enzymes

The immobilization of enzymes to carrier materials is carried out with the intention of minimizing autolysis while ensuring a high enzyme-to-substrate ratio. The large surface density of the enzyme on the support matrix and the high S/V ratio of the solid support provide the means of achieving increased availability of the enzyme to the substrate. A beneficial feature of immobilizing enzymes is the possibility for its simple reuse without its isolation from the post-reaction mixtures. Immobilization minimizes enzyme consumption and allows for obtaining a higher product yield. Furthermore, enzyme stability can also be increased [11].

The immobilization process can be performed either off-site (e.g., when the enzymes are attached to the surface of beads outside the μ-IMER and then a micropacking is formed from the beads in the microchannel) or in-site, when the enzymes are immobilized directly onto the internal surface of the reactor.

### 3.1. Modes of Immobilization

The strategy used for enzyme attachment can have a huge impact on proteolytic performance, lifetime and reusability, therefore the proper choice of immobilization chemistry is of substantial importance. Several techniques have been developed so far, including adsorption, covalent bonding, bioaffinity interaction, entrapment/encapsulation and the cross-linking of enzymes (Figure 4). Herein, we provide a short overview of the methods that have been utilized in recent contributions (2017–2021, Table 2); for a more comprehensive description, the reader is kindly referred to previous reviews [38,71,72].

#### 3.1.1. Adsorption

The simplest approach is probably when the enzyme is attached to the surface by weak, intermolecular forces, such as hydrogen bonding, electrostatic forces and hydrophobic interactions. Enzyme molecules are therefore not subjected to chemical modifications. Kecskemeti et al., developed a straightforward technique that takes advantage of the hydrophobic nature of PDMS for the spontaneous adsorption of trypsin [13]. Upon contact with the unmodified PDMS microchannels, the hydrophobic side chains of the enzyme arrange towards the hydrophobic surface. The limited lifetime (~2 h) of this μ-IMER was also highlighted, which was attributed to enzyme unfolding on the PDMS surface. Such spreading phenomena are thought to be the result of the continuous conformational rearrangement of the enzyme. Since the trypsin is in direct contact with the PDMS, the transition between these conformational states easily exposes an increasing number of hydrophobic side chains on the hydrophobic surface [13,61]. In theory, the use of a spacer molecule could alleviate this problem. The very easy and fast regeneration of the reactor, however, makes up for its limited lifetime.

Another technique based on adsorption is the electrostatic interaction between the oppositely charged enzyme and the surface. In such cases, special attention should be paid to the pI value of the enzyme. Trypsin (pI ~ 10.3) was successfully immobilized on a fused silica capillary, taking advantage of the charged state difference in a relatively wide pH range [48]. Trypsin, having a net positive charge below its pI readily binds to the negatively charged capillary surface (silanol groups deprotonate above pH ~ 3). Using this trail of logic, it would not be possible to achieve the direct immobilization of, e.g., pepsin due to its low isoelectric point (pI ~ 3).

Recently, Zhang et al., reported on a novel method for trypsin immobilization based on electrostatic interaction [67,68]. Graphene-oxide (GO)-modified microparticles served as a solid support, which proved to accommodate a larger amount of trypsin than the microparticles not modified with GO nanosheets used in a previous study [67,80].

The main problems associated with adsorption-based immobilization (activity loss, enzyme leaching) can be solved by establishing a more stable bond between the enzyme and the solid support (e.g., covalent- or bioaffinity linkage).

#### 3.1.2. Covalent Coupling

Immobilization through covalent linkage is the most widely used approach, where enzymes are attached to the surface functionalities via their amino or thiol groups. This typically necessitates the incorporation of a multi-step pre-treatment procedure in order to render the surface of the carrier material reactive. The popularity of covalent coupling stems from its innate ability to form stable bonds, suppressing enzyme leaching, which can generously prolong the lifetime of the μ-IMER (~months). One of the main concerns, however, is the fact that all available amino or thiol functionalities in the protein molecule (including those of the active site) can be exposed to the reactive groups of the surface, therefore there is a danger of activity loss. In order to restrain such behavior, a reversible enzyme inhibitor (e.g., benzamidine for trypsin [53,56]) can be used to block the active site from participating in the immobilization. Another reason for reduced activity despite the high enzyme load can be the limited accessibility of the enzyme active site mainly due to its arbitrary orientation on the surface. Increasing the distance between the enzyme and the support through a spacer molecule can improve steric access [55].

In a recently published paper, a novel method was proposed, where the zymogen form of trypsin—trypsinogen—was immobilized. The surface-bound trypsinogen was converted to active trypsin by enterokinase prior to use. Trypsinogen immobilization was compared to conventional trypsin immobilization, and it was found that the latter yielded considerably lower enzyme density on the surface, which was explained by the detrimental effect of autoproteolytic processes occurring during trypsin immobilization [59].

A diverse arsenal of reaction schemes is available for anchoring the enzymes on the surface. One of the most frequently utilized techniques is glutaraldehyde coupling [47,52,59]. Aldehyde groups are grafted onto the surface, which can then react with the amino groups present in the enzyme. The resulting imine (Schiff base) can be further reduced (e.g., with NaCNBH_3_) to an amine. Another strategy that has a long history is based on carbodiimide cross-linking [16,53,60]. Here, a stable amid bond is formed between activated carboxyl groups and amine groups. Contrary to carbodiimides, activation by carbonyldiimidazole can be carried out in non-aqueous media for the conjugation of surface hydroxyls or carboxyls and the amine groups present in the enzyme [55]. The utilization of azlactone chemistry has also proved valuable for enzyme immobilization [56,65,66]. Azlactone moieties readily react with the amine and thiol groups of biomolecules via a ring-opening reaction [81]. Such straightforward reactions can also be accomplished with thiol-ene (TE) click-chemistry [50,54,57,58]. The basis of such TE-driven immobilization is the reaction between thiol and alkene functionality. Support materials containing double bonds on their surface easily react with the free thiol groups of enzymes, forming a thioether bond. In such cases, enzymes are pre-treated with reducing agents to generate free thiol groups from disulfide bridges. In addition to double bonds, the free thiol groups of the enzyme can also interact with gold nanoparticles located on the surface [51,62]. This thiol-gold interaction results in the formation of a dative bond (Au-S).

#### 3.1.3. Bioaffinity Linkage

The previously described methods (adsorption, covalent coupling) lack the orientational control of enzyme immobilization, which at times can lead to unanticipated activity loss. This is when bioaffinity linkage comes into the picture. Bioaffinity immobilization exploits the highly specific interaction between affinity pairs, e.g., antigen-antibody, biotin-avidin, providing stable and oriented attachment.

Recently, a method employing DNA-directed immobilization (DDI) was demonstrated [49]. This technique required the functionalization of the support surface with single-strand DNA (ssDNA) and the precoupling of trypsin to the complementary DNA strand. Immobilization occurred via DNA hybridization, whereby the two single strands of DNA were annealed to each other. The authors also presented proof of increased stability compared to a μ-IMER utilizing covalently bound trypsin.

### 3.2. Supports for the Immobilization

The immobilization of an enzyme onto high surface-to-volume ratio support reduces the diffusion length which promotes the convergence of substrate molecules to the active sites of immobilized enzymes. The main types of enzyme reactors are (1) wall-coated/open tubular (including micropatterned microchannels), (2) packed-bed and (3) monolithic microreactors. There are several other solid supports with a high specific surface area, such as membranes, papers or gels, which only slightly differ from the approaches mentioned above.

#### 3.2.1. Open Tubular Channel/Capillary

The wall-coated configuration of μ-IMERs represents a reactor where the enzyme is immobilized on the surface of the inner wall of the microfluidic channel or capillary, forming a catalytic layer. From among the three basic reactor types, the open tubular (OT) systems possess the lowest S/V ratio, which imposes constraints on enzyme-loading capacity. Furthermore, the low S/V value also implies a relatively long diffusion path between the substrate and surface-bound enzyme.

The diffusion distance can be reduced by incorporating microstructures (e.g., pillars [14,62]) into the channel. A significant increase in S/V ratio can be expected in tightly patterned microchannels, however, the technique used for microfabrication can have its limitations regarding miniaturization [14]. The manipulation of fluid flow also facilitates diffusion-limited mass transfer. Despite the strongly laminar flow conditions in microchannels, the integration of curvatures in the channel design can promote the movement of substrate molecules by inducing a spiraling fluid flow [73].

Lengthening the channel and/or allowing longer contact time can be a reasonable approach to enhance reactor efficiency. Increasing reactor length is analogous to applying a parallel channel system. Parallelization can either be achieved in chips using microfabrication techniques or by the utilization of commercially available multi-lumen capillaries (MLC) [66,70]. Such MLCs are typical of the same dimension as conventional fused silica capillaries, only they contain an array of microchannels (inner diameter <10 μm), therefore its S/V ratio is markedly enhanced (Figure 5).

When OT configurations with improved geometry do not yield the desired effect, one can also resort to the modification of the inner surface, either by depositing a porous layer [47,66] or by introducing nano-architectures (e.g., GNPs [62,70]). The main argument in favor of OT platforms is the practically negligible backpressure they produce and their relatively straightforward preparation.

#### 3.2.2. Packed Channels

This configuration of μ-IMERs resembles traditional packed-bed reactors or chromatographic (LC) or solid-phase extraction (SPE) microcolumns. The enzymes are pre-immobilized onto solid supports, which are then integrated into a microchannel or capillary. The specific surface area of the reactor can be tuned by employing particles of varying size (micro- and nanoparticles) and porosity (fully-, superficially or non-porous). Remarkably high S/V and enzyme load can be achieved with decreasing particle size and increasing porosity, however, the presence of pores is the very reason why the full exploitation of the extremely high enzyme-to-substrate ratio offered by these platforms cannot be exercised. Naturally, densely packed channels greatly reduce the diffusion distance, however, the theoretically attainable mass transfer rates are reduced due to the time-scale of substrate partitioning in and out of the pores [81].

Another challenge from a practical point of view is the packing procedure with a specific emphasis on particle retention. Obstructions have to be integrated into the conduit to hold the particles in place. For such purposes, frits are often applied. These can either be manually adapted to the reactor ends [69,82] or in situ formed by chemical reaction [76]. In the case of chips—since one has considerable control over channel architecture—channel structures promoting particle retention can be designed. Typically, tapering or the integration of bottlenecks [16] are utilized. This way, the manual or chemical maneuvering required for particle retention in capillary-based approaches can be circumvented.

Beads/particles come in a variety of materials—silica [16], cellulose [69], polymers, e.g., polystyrene divinylbenzene (PSDVB) [76] or acrylate [67,68]. Functionalization with proteases can be performed in-house utilizing the well-known immobilization strategies, such as covalent bonding [16] and adsorption [67]. In addition, there are commercially available trypsin-modified beads (e.g., Poroszyme^®^ [76]).

The integration of magnetic particles is quite unique and brings several advantages to the table. First of all, the positioning of the particles is carried out with an external magnetic field, therefore the integration of mechanical barriers is not a necessity. Magnetic particles can easily be recycled, since the use of external magnets enables the removal of these particles, as well. Bataille and coworkers used magnetic beads (diameter: 2.8 μm) to create a magnetic micro fluidized bed [60]. In this system, the experimental conditions were developed such that the packed-bed acquired fluidic properties [83]. As a result of this, interparticle distances increase and particles become mobile, which can maximize enzyme-substrate interactions.

#### 3.2.3. Monoliths

In the monolithic μ-IMERs the microchannels are formed by the interconnected meso- and macro-porosity of the material. Monoliths enjoy increasing popularity, as these can combine the advantageous features of both open tubular and packed-bed reactors, offering a remedy for the shortcomings typical of the latter two supports. The pores provide an elevated S/V ratio relative to wall-coated reactors as well as lower backpressure and higher mass transfer rates compared to the particle-based configuration. The material of the monoliths can be inorganic (e.g., silica), organic or hybrid. Predominantly, organic polymer monoliths are used for μ-IMER applications due to their high resistance to extreme pH conditions, straightforward preparation and ease of modification with functional groups. Their synthesis requires the following ingredients: monomers, (initiator), crosslinker and porogen. The porogen is a key component responsible for forming the porous structure. The choice of its composition and the monomer-to-porogen ratio has a huge impact on the overall morphology and permeability of the monolith.

Polymer monoliths based on methacrylate monomers are widely employed [52,54,55,57,58]. After polymerization, the residual double bonds on the surface can serve as direct attachment points for the enzyme (TE click-chemistry) [54,58] (Figure 6). Surface properties can be optimized by the choice of monomers. Hydrophilic monomers can be used in order to render the surface less susceptible to non-specific adsorption phenomena [55]. Surface functionalities can also be tuned by utilizing monomers having appropriate functional groups for the desired immobilization chemistry, e.g., the incorporation of azlactone monomers for ring-opening reactions [56]. Very recently a macroporous monolith based on high internal phase emulsion (HIPE) was developed. The macropores yielded higher permeability, although at the expense of a lower S/V ratio. The S/V was therefore increased by surface functionalization with gold nanorods [51].

The application of monolithic supports utilizing thiol–ene (TE) chemistry seems to be flourishing, especially in microchip-based implementations. In the recently published works, these monoliths were prepared almost exclusively in TE chips, creating all thiol-ene μ-IMER devices. The channel system of a TE chip proves to be exceptionally beneficial for housing such TE monolithic beds since the material is adhered to itself, which yields well-anchored monoliths. By using the “thiol” and “ene” components in an off-stoichiometric ratio, the resulting surface can be furnished with either sulphydryl or allyl functionalities, providing a suitable environment for enzyme immobilization. The versatility of the system is demonstrated by the fact that a variety of enzymes (trypsin [60], pepsin [63,64], PNGase [75]) could be immobilized without carrying out further optimization steps specific to the enzyme.

#### 3.2.4. Other Supports

Although in the present review the focus of attention is on proteolytic μ-IMERs, there are interesting examples in the literature utilizing other supports or configurations that, in a strict sense, may not fall into the category of a micro fluidic reactor. In these cases, the volume and/or the dimensions of IMERs were larger than the microscale.

The preparation of a porous PDMS monolith was demonstrated by Liu et al. [84], wherein a bed of glass microbubbles (average diameter: 55 μm) was used as a sacrificial template for creating the monolithic framework. The porous structure was formed by etching away the microbubbles from the PDMS-microbubbles composite (average pore size: 51 μm). The resulting monolith was placed in a plastic column and a syringe pump was used to introduce sample solutions.

The use of porous membranes fixed in membrane holders is also a popular choice [77,79]. Trypsin was immobilized on commercially available nanoporous anodized alumina membranes, creating a flow-through IMER that can be on-line coupled to ESI-TOF-MS [79]. The developed IMER was characterized by a high trypsin load due to the nanopores (pore size: 200 nm), as well as enhanced stability and straightforward assembly. In another study, nylon membranes with varying pore sizes (0.45, 1.2 and 5 μm) were used as supports for immobilizing trypsin both covalently and electrostatically [77]. The authors investigated the effect of immobilization strategy and increasing pore size on digestion efficiency.

Enzymes have also been attached to porous ceramic capillary membranes [85,86]. The tubular shape with an inner diameter of ~ 1 mm was prepared by extrusion. Ceramic materials possess several appealing features, such as high chemical, thermal and mechanical resistance as well as tunable pore structure and geometry, making them an attractive choice as supports for IMER.

## 4. Proteomic Applications

The inherent advantages of microfluidic IMERs can be well exploited in proteomics, especially in bottom-up workflows. By immobilizing proteases, their autolysis, which generally decreases their activity and contaminates the sample with peptides characteristic of the enzyme, can be minimized. Furthermore, immobilization can result in increased thermal stability and organic solvent resistance of enzymes.

Trypsin is by far the most often used proteolytic enzyme owing to its high specificity. It hydrolyzes the C-terminal peptide bonds of arginine and lysine residues. Since proteins are usually rich in such amino acids, the peptides generated by proteolysis are typical of a size that is sequenceable by tandem mass spectrometry, but still specific to the protein. In some cases, trypsin treated with N-tosyl-L-phenylalanine chloromethyl ketone (TPCK-trypsin) was used to avert any extraneous chymotryptic activity [47,56,59,60,67].

Although most publications present the use of tryptic IMERs, other enzymes are utilized, as well. Pepsin is also a popular enzyme; however, its specificity is lower than that of trypsin—the preferential cleave sites are at the hydrophobic, aromatic residues. Monolithic pepsin μ-IMERs were used for digesting hemoglobin [64] and for inhibitor screening of natural products [52]. In another study utilizing pepsin, the authors demonstrated the integration of a microreactor with micro free-flow electrophoresis for the first time [63]. To improve the efficiency of proteolysis, dual enzyme reactors can be used. Lys-C and trypsin were co-immobilized covalently in a multi-channel reactor, which was on-line coupled to a nanoLC-MS system. Ricin extracted from castor beans was identified by signature peptides after a 5-min long digestion time. The generation of these peptides did not require the reduction and alkylation of the sample [66]. An α-chymotrypsin IMER was created in a TE microfluidic chip functionalized with GNPs. Bradykinin (an ideal peptide for chymotryptic digestion) was used as a model substrate for monitoring the activity of the μ-IMER [62].

Proteomic μ-IMER-related publications often focus on the fabrication of the microfluidic device and the immobilization of the enzyme. The resulting IMER is then characterized with simple substrates, peptides or protein standards. Additionally, complex biological samples are also used to demonstrate the suitability of the IMER for real-life applications. The protein content of mouse [54,58] and rat liver [51,57], snake venom [14] as well as extracts from Escherichia coli cells [67,68,69] have been analyzed. A method for ricin detection was developed using castor seeds [66]. Clinical samples like plasma [79], serum [16], tear [48,74], saliva [73] or dried blood spot samples [65] are popular due to their potential medical significance (e.g., biomarker discovery). Different types of cancer cells, namely hepatocarcinoma [67,68], breast cancer (MCF-7) [50], and HeLa cells [47,76] were analyzed.

Wei et al., integrated the most time-consuming steps of bottom-up proteomic analyzes into a glass microfluidic chip. Reduction and alkylation were carried out consecutively in serpentine channels by injecting the protein, DTT and IAA solutions simultaneously into the appropriate chip channels (Figure 7). Thiol-ene click-reaction was utilized for the immobilization of trypsin. Using this chip for the analysis of protein extract from mouse liver, a large number of peptides and proteins with a wide range of molecular weights and isoelectric points could be identified using HPLC-MS/MS while reducing the digestion time by 66 times compared to the conventional in-solution procedure [58]. In subsequent work, a fused silica capillary containing a monolith was connected to the protein inlet port of the chip enabling protein fractionation. The outlet of the tryptic μ-IMER was connected to another monolith, which was imprinted with a tripeptide characteristic of ubiquitin-modified proteins, facilitating the selective enrichment of these characteristic peptides. This system enabled a 2.8-fold increase in the number of identified proteins in breast cancer cells (MCF-7) compared to the untreated sample [50].

The integration of enzymatic digestion and separation into a microfluidic chip was implemented by Lu and colleagues. The chip was fabricated from the thiol-ene polymer using the double-molding technique, and it was designed for free-flow electrophoresis with multiple inlet and outlet channels and partitioning bars between the separation chamber and electrode channels to prevent bubbles from entering the main chamber. The separation capabilities were characterized with fluorescent dyes and fluorescein isothiocyanate labeled amino acids. In the main inlet channel, an μ-IMER was formed by covalently immobilizing pepsin onto an in situ created thiol-ene monolith. Glu-fibrinopeptide was used as a model substrate to test the in-line digestion and separation [63].

Single-cell analyses are becoming more significant, but in the case of single-cell shotgun proteomics, bulk methods would result in immense dilution. Considering that the volume of a single cell is on the scale of picoliters, the volume of typical microreactors (microliters to nanoliters) is still too large. A microreactor with nanochannels was designed by Yamamoto et al., to use as picoliter IMER. E-beam lithography and dry etching techniques were used to create these channels in a fused silica glass substrate. Trypsinogen was immobilized in the channels using glutaraldehyde, which was then activated with enterokinase. Using this method, the trypsin concentration was 2.5 times higher than with the typical method of immobilizing the active trypsin directly [59]. Besides dilution, sample loss during preparation and inefficient digestion are also potential issues of single-cell proteomics. Addressing these challenges, Hata et al., developed a method called in-line sample preparation for efficient cellular proteomics (ISPEC), which was tested and optimized using HeLa cells ranging from a thousand to a single cell. Using a nanosyringe pump, first the lysis solution, and then the cells were aspirated into a sampling capillary with the aid of a micromanipulator and a microscope. The cells and the lysis solution were mixed by flowing water at a low rate through the capillary. Proteins from the cells were transferred to a second capillary, which was packed with commercially available immobilized trypsin beads. The generated peptides were trapped at the top of a nanoLC column and desalted using an appropriate mobile phase before nanoLC/MS/MS analysis. The method helped minimize sample loss, optimize digestion, and was able to identify 60 proteins from a single cell [76].

Quantitative proteomic techniques rely predominantly on isotope labeling. Proteolytic ^18^O labeling is a common strategy utilizing H_2_^18^O as media for enzymatic digestion, incorporating digestion and isotope labeling into a single step, and eliminating the need for extra labeling reagents. Incomplete digestion is the main possible issue with this method. For this reason, Yuan et al., created an ultra-performance capillary-based μ-IMER, in which TPCK-trypsin was immobilized onto polyetherimide-modified acrylic polymer microspheres covered with graphene-oxide nanosheets. The μ-IMER was on-line coupled to nanoHPLC-MS, and protein extracts from Escherichia coli were analyzed. In a 2.5-min reaction time, a 99% labeling efficiency was achieved with only 8% of the peptides containing missed cleavages [67]. The same research group used on-line dimethyl labeling in later work. An μ-IMER was prepared the same way as previously, and the digests were captured in a C18 trap column. For isotope labeling, light and heavy isotope labeling reagent were pumped through the trap column. The on-line digestion and isotope labeling system was supplemented with an SCX column before the RP separation column, making multidimensional separation possible, while shortening these steps from 20 h to 3 h. Labeling and digestion were carried out in a similar manner as in previously published work. Both systems were used for the large-scale relative proteome quantification of hepatocarcinoma cells extracts [68].

Hydrogen/deuterium exchange mass spectrometry (HDX-MS) can be utilized in protein conformation, dynamics and interactions studies, but PTMs, such as disulfide bonds and glycosylations can present challenges. Comamala and colleagues immobilized PNGase A, PNGase H+ and PNGase Dj enzymes (the latter two were produced by transfected E. coli) covalently on TE monolith created in TE chip channels, and tested them with HILIC-enriched Trastuzumab tryptic digest under quench conditions (pH 2.5 and 0 °C). Electrochemical reduction of disulfide bonds using a commercial μ-PrepCell SS device (Antec Scientific) yielded a 96% reduction when tested with an insulin-based system, and pepsin μ-IMER provided an almost complete sequence coverage of hemoglobin. Electrochemical cell, pepsin and PNGase μ-IMERs were online connected to a UPLC-HDX-MS setup, and the epitope mapping of a mAb to the sema domain of the tyrosine-protein kinase Met (SD c-Met) in its native form was carried out. This method showed improvement in effective sequence coverage compared to the conventional method, but also caused an increase in back-exchange. Back-exchange could be decreased using a higher flow, possibly sacrificing deglycosylation efficiency [75].

## 5. Conclusions

The μ-IMER technology has been developing intensively and is being applied in many different industrial and scientific research areas. Furthermore, μ-IMERs are often prepared in a relatively low-flow format, such as a capillary. However, if the inner diameter of these capillaries reaches 1 mm (e.g., [85,86,87]) then these devices cannot be considered microfluidic systems. μ-IMERs are generally very small devices/chips consisting of channels/capillaries with inner diameters less than a few hundred micrometers. These conduits can contain particles, monoliths and porous membranes, as well. The last decades have seen the emergence of several other possible solid supports used in microfluidic chips, however, these were not applied as proteomic μ-IMERs in the last five years (e.g., paper-based chips [88,89]). Beyond the materials mentioned previously, metal-organic frameworks (MOFs) are a relatively new addition to the family of support matrices for immobilization [90]. So far, MOFs have not been utilized for proteomic-related studies extensively, however, they hold great potential for such purposes.

The confinement of enzymes to solid supports in a microenvironment is advantageous due to the high surface-to-volume ratio offered by such systems. However, several factors have to be taken into consideration for the exploitation of the high enzyme-to-substrate ratio. The choice of immobilization strategy, as well as support material/configuration, has a huge impact on the overall μ-IMER performance. Although enhancing the specific surface area is a popular approach to promote enzyme-substrate interaction, the increased enzyme load can also result in restricted access to the active site of the enzyme.

The incorporation of μ-IMER devices into bottom-up proteomic workflows (either off-line, on-line or in-line) certainly has its advantages. These microreactors enable accelerated digestions, enzyme reusability and low sample consumption and can even allow the circumvention of reduction and alkylation, which are also time-consuming steps [65] (however, these can only be omitted when proteins containing relatively few S-S bridges are analyzed).

There has been a tremendous improvement regarding immobilization strategies and the choice of solid supports. A handful of μ-IMERs showing excellent performance have been developed so far. In order to establish genuinely high throughput, fully automated systems and the on-line integration of μ-IMERs is inevitable. The development of an on-line platform is quite a demanding task since the experimental conditions for proteolysis, subsequent separation and MS detection need to be adjusted such that compatibility is achieved between each unit. Several works have addressed these challenges and demonstrated on-line coupling, with some of them utilizing custom-designed fittings/3D printed interfaces. In the future, endeavors are likely to shift towards the development of μ-IMERs that allow on-line integration.

## Figures and Tables

**Figure 1 micromachines-13-00311-f001:**
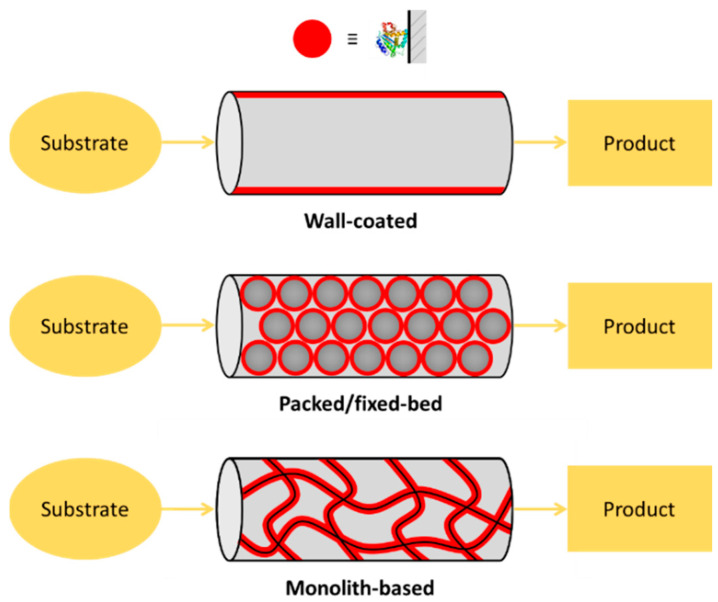
Types of enzyme-immobilized microreactors. (1) Wall-coated enzyme-immobilized microreactor, where the enzyme is directly adsorbed onto the inner surface of the microchannels or capillary. (2) Packed/fixed-bed enzyme microreactor, where the enzyme is pre-immobilized into particles/beads that are packed. (3) Monolithic microreactor, where the enzyme is immobilized onto the surface of the pores/channels of a monolithic material.

**Figure 2 micromachines-13-00311-f002:**
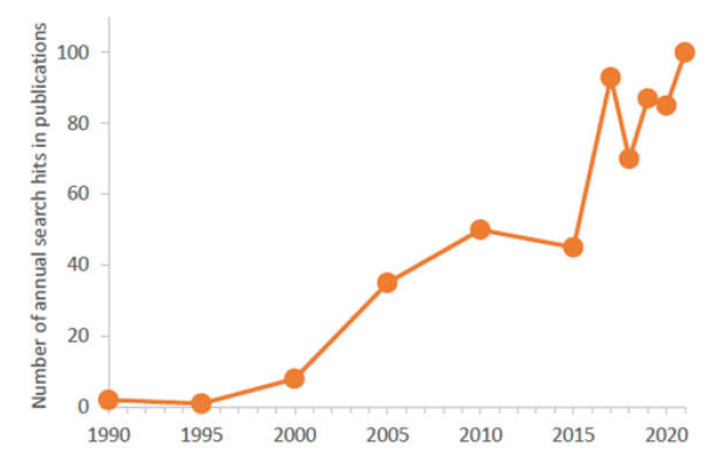
Number of annual search hits related to “MICROFLUIDIC ENZYME REACTOR” (searching with Google Scholar) where the keywords appeared in the articles.

**Figure 3 micromachines-13-00311-f003:**
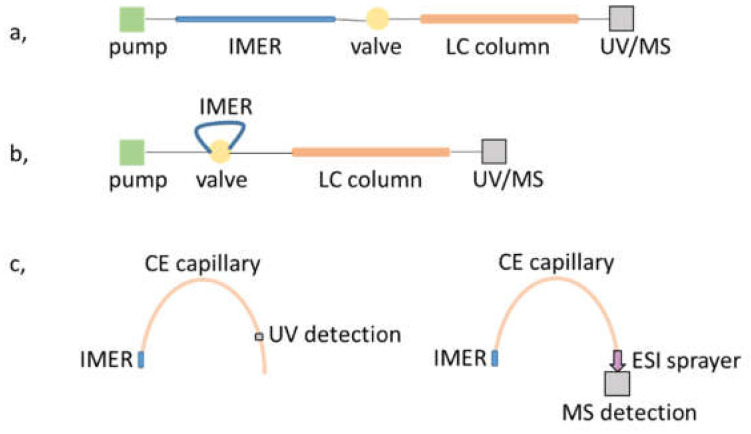
Schematic drawing of IMER-separation unit workflows for (**a**,**b**) LC and (**c**) CE platforms with UV and MS detection.

**Figure 4 micromachines-13-00311-f004:**
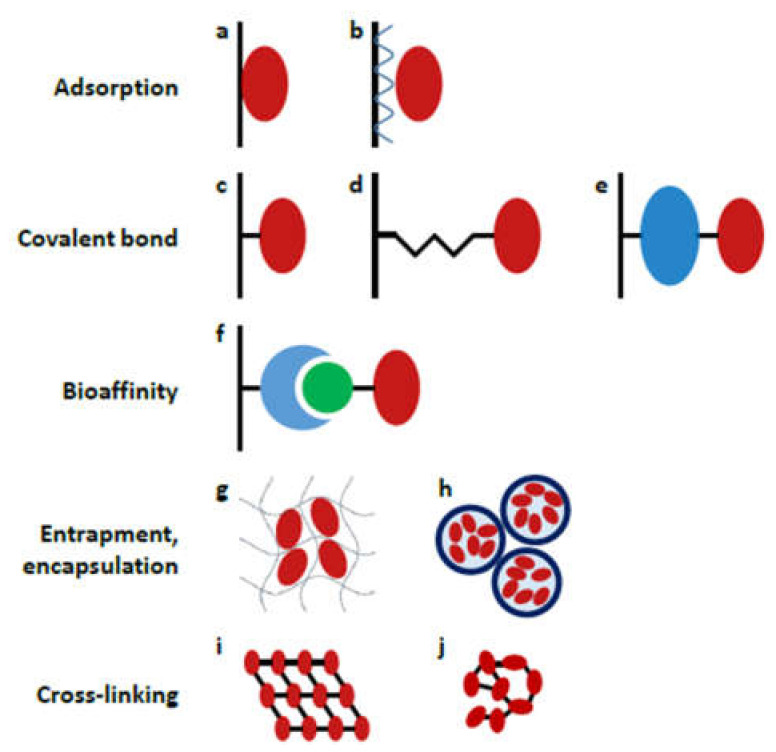
Schematic representation of different enzyme immobilization strategies: (**a**) direct adsorption, (**b**) layer-by-layer adsorption; covalent coupling (**c**) directly (e.g., EDC/NHS coupling), (**d**) through a short spacer (e.g., glutaraldehyde), (**e**) through a large spacer (e.g., albumin); (**f**) coupling by bioaffinity interaction (e.g., avidin-biotin interaction); (**g**) entrapment into a gel matrix, (**h**) encapsulation into polyelectrolyte capsules; cross-linking to form (**i**) ordered crystals or (**j**) unordered aggregates. (Reprinted with permission from [38], published by Elsevier (Amsterdam, The Netherlands), 2018.).

**Figure 5 micromachines-13-00311-f005:**
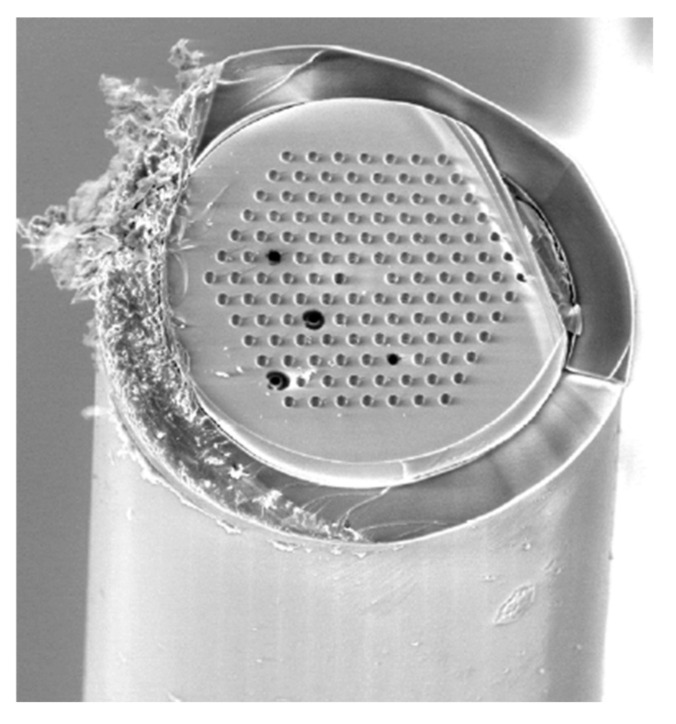
Scanning electron micrograph of MLC containing 126 parallel channels. (Reprinted with permission from [66], published by the American Chemical Society (Washington, DC, USA), 2017).

**Figure 6 micromachines-13-00311-f006:**
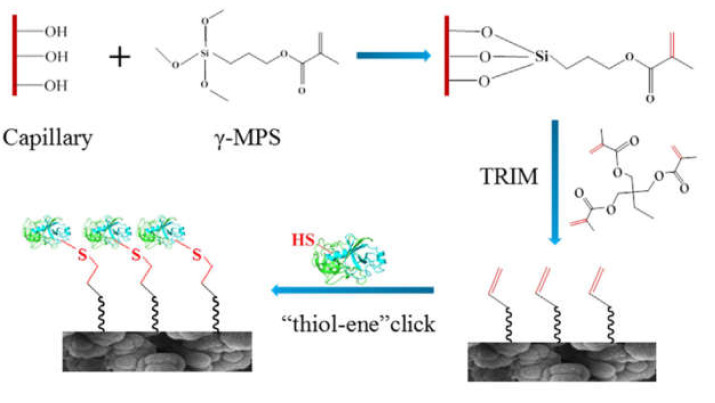
Reaction scheme of preparing a μ-IMER in a fused silica capillary containing methacrylate-based monolith. (Reprinted with permission from [54], published by Elsevier, 2020).

**Figure 7 micromachines-13-00311-f007:**
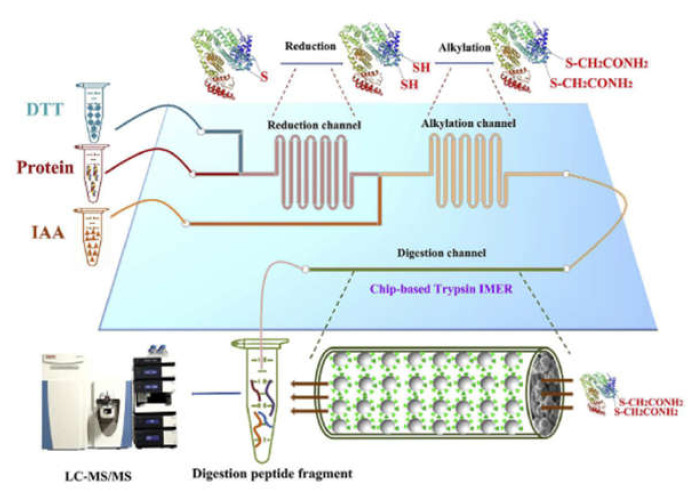
Schematic illustration of the integration of chip denaturation—chip IMER digestion. (Reprinted with permission from [58], published by Elsevier, 2020).

**Table 1 micromachines-13-00311-t001:** Recent reviews of immobilized enzymatic reactors (2017–2021).

Ref.	Title	Keywords (max. 4) *
[35]	Characterization and evaluation of immobilized enzymes for applications in flow reactors	biocatalysis, protein immobilization, advanced materials, packed-bed reactors
[7]	Recent developments in microreactor technology for biocatalysis applications	enzymatic microreactor; biocatalysis; monolith; multiphase systems
[36]	Review on membranes in microfluidics	membranes; manufacturing methods; applications; mass transfer
[37]	Magnetic microreactors with immobilized enzymes—from assemblage to contemporary applications	enzymatic microreactors; magnetic particles; nanomaterials; immobilization
[38]	Particle-based immobilized enzymatic reactors in microfluidic chips	enzyme reactor; particle; enzyme immobilization; protein digestion
[39]	Catalytic membrane microreactors for fuel and biofuel processing	membrane; catalytic membrane microreactors; microchannels; catalytic processes
[10]	Biocatalysis in continuous-flow microfluidic reactors	enzyme immobilization; flow biocatalysis; microfluidic reactors; miniaturization
[11]	Microfluidic reactors with immobilized enzymes—characterization, dividing, perspectives	immobilized enzyme microreactor; miniaturization; immobilization strategies; biocatalysis
[40]	Immobilized enzyme-based analytical tools in the -omics era: recent advances	immobilized enzyme reactors; proteomics; glycomics; dual IMERs
[41]	Recent progress of microfluidic reactors for biomedical applications	microreactor; PCR; ELISA; hybridization
[42]	Immobilized enzyme reactors integrated into analytical platforms: recent advances and challenges	hyphenation; enzymatic reaction; immobilization; liquid chromatography
[43]	Microfluidic reactor with immobilized enzyme—from construction to applications	microfluidic IMER; immobilization strategies; biocatalysis; bioconversion
[44]	Microfluidic immobilized enzyme reactors for continuous biocatalysis	in vitro biocatalysis; microfluidic reactor; enzyme immobilization; multi-enzyme systems
[45]	On-line microfluidic immobilized enzyme reactors: A new tool for characterizing synthetic polymers	biodegradable polymer; enzymatic degradation; polyesters; lipase
[21]	Enzyme embedded microfluidic paper-based analytic device (μPAD): a comprehensive review	microfluidic devices; hybrid nanoflowers; design and fabrication; point-of-care

* The keywords were obtained from the given paper.

**Table 2 micromachines-13-00311-t002:** Comparison of the μ-IMERs applied for proteomics published between 2017–2021.

Immobilized Enzyme	Reactor Type	Type of Solid Support	Enzyme Immobilization Strategy	Coupled Detector-Analyzer	Application	Ref.
trypsin	fused silica capillary	monolith	TE click-reaction	LC-MS	protein extract digestion, breast cancer (MCF-7) cells	[50]
trypsin	glass microchip, fused silica capillary	monolith	TE click-reaction	LC-MS	protein extract digestion, mouse liver	[58]
TPCK-trypsin	PDMS chip (microfluidized bed) TE microchip	magnetic bead monolith	covalent (carbodiimide) TE click-reaction	LC-MS	protein standard digestion	[60]
pepsin	TE microchip	monolith	TE click-reaction	SDS-PAGE, LC-MS	protein standard digestion	[64]
trypsin	PDMS microchip	channel wall	adsorption	CE-UV, LC-MS	protein standard digestion	[13]
trypsin	PDMS microchip	silica particles	covalent (carbodiimide)	CE-UV, LC-MS	protein extract digestion, human serum	[16]
pepsin	TE microchip	monolith	TE click-reaction	FFE, ESI-MS	peptide digestion	[63]
α-chymotrypsin	TE microchip	GNPs	thiol-gold interaction	ESI-MS	peptide digestion	[62]
trypsin	COC microchip	monolith	covalent (azlactone chemistry)	nanoLC-MS	protein extract digestion, dried blood spots	[65]
TPCK-trypsintrypsinogen	glass microchip	derivatized channel wall	covalent (glutaraldehyde)		substrate digestion	[59]
trypsin	PDMS microchip	channel wall	adsorption	CE-UV, CE-MS	protein extract digestion, snake venom	[14]
trypsin	PDMS microchip	channel wall	adsorption	CE-UV, CE-MS	protein extract digestion, saliva	[73]
trypsin	PDMS microchip	channel wall	adsorption	CE-UV, CE-MS	protein extract digestion, tear	[74]
PNGase A, Dj, H+	TE microchip	monolith	TE click-reaction	LC-HDX-MS	deglycosylation	[75]
trypsin	fused silica capillary	PSDVB particles	commercial immobilized beads (covalent)	nanoLC-MS	protein extract digestion, HeLa cells	[76]
trypsin	capillary	GNR- functional-ized monolith	thiol-gold interaction	nanoLC-MS	protein extract digestion, rat liver	[51]
pepsin	capillary	polymer monolith	covalent (glutaraldehyde)	CE-UV	substrate digestion, inhibitor screening	[52]
trypsin/Lys-C	MCR	polymer layer	covalent (azlactone chemistry)	nanoLC-MS Q-Ex	protein extract digestion, castor bean	[66]
trypsin	capillary	polymer monolith	covalent	CE-UV, HPLC-UV	protein standard digestion	[53]
cathepsin D	capillary	derivatized channel wall	covalent (glutaraldehyde)	HPLC-FD	peptide digestion, inhibitor screening	[46]
trypsin	MCR	GNPs	covalent	capLC-UV, nanoLC-MS	protein standard digestion	[70]
trypsin	capillary	monolith	TE click-reaction	nanoLC-MS	protein extract digestion, egg white, mouse liver	[54]
TPCK-trypsin	capillary	porous layer	covalent (glutaraldehyde)	nanoLC-MS	protein extract digestion, HeLa cells	[47]
trypsin	capillary	polymer monolith	covalent	nanoLC-UV, MALDI-TOF MS	protein standard digestion	[55]
trypsin	capillary	channel wall	adsorption	CE-UV, CE-MS	protein extract digestion, tear	[48]
TPCK-trypsin	capillary	polymer monolith	covalent	LC-UV	protein standard digestion	[56]
trypsin	capillary	cellulose resin	commercial immobilized beads	CE-MS	protein extract digestion, E. coil	[69]
trypsin	capillary	channel wall	DNA-directed	CE-UV, MALDI-TOF MS	protein standard digestion	[49]
TPCK-trypsin	capillary	GO-modified polymer microsphere	electrostatic interaction	nanoLC-MS, MALDI-TOF MS	protein extract digestion, E. coil, Hca-F and Hca-P cells	[67]
trypsin	capillary	GO-modified polymer microsphere	electrostatic interaction	2D nanoLC-MS, MALDI-TOF MS	protein extract digestion, E. coil, Hca-F and Hca-P cells	[68]
trypsin	capillary	monolith	TE click-reaction	CE-UV	protein extract digestion, rat liver	[57]
trypsin	membrane holder	porous membrane	adsorption, covalent (carbodiimide)	UV, SDS-PAGE, ESI-MS	protein standard digestion	[77]
trypsin	capillary	monolith	covalent	LC-UV, LC-MS	protein standard mixture digestion	[78]
trypsin	membrane holder	nanoporous alumina membrane	covalent (CDI)	SDS-PAGE, MALDI-TOF MS, ESI-MS, nanoLC-MS	protein extract digestion, human plasma	[79]

## Data Availability

Not applicable.

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
