# Peer review of "Microfluidic Immobilized Enzymatic Reactors for Proteomic Analyses—Recent Developments and Trends (2017–2021)"

_micromachines, 2022, doi:10.3390/mi13020311_

Round 1

Reviewer 1 Report

Review DOI: 10.1039/D0RE00483A provides some metrics for microreactors, a set of parameters that is missing in the present work, information on this matter would be of interest for this review

Review DOI: 10.1007/978-1-0716-0215-7_16 focuses on coated wall microreactors information gathered there should be considered in the present work

Immobilization of multiple enzymes in microreactor for implementation of cascade reactions, should be considered, see, e.g., https://doi.org/10.1002/biot.201700030. For strategies for multi-enzyme immobilization see e.g., https://doi.org/10.3389/fbioe.2020.00660

The use of novel immobilization methods, e.g., metal organic frameworks should be addressed, see e.g., https://doi.org/10.1016/j.ccr.2019.213149

Reviewer 2 Report

In this review, the authors summarize the recent developments of microfluidic immobilized enzymatic reactors (µ-IMER) and their use in proteomics. The review sequentially discusses the recent advances in IMER fabrication, strategies for enzyme immobilization, and finally focus on the recent advances in proteomic analyses and some perspectives. Overall, this review is well-organized and has good communication. It would be instructive and informative to readers who want to quickly learn about the very recent developments in the IMER field and how it works for proteomics. Besides, it has clear and well-organized summaries of all the related IMER review papers and µ-IMER for proteomic papers in Tables 1 and 2 for extensive readings. I would recommend its publication after addressing some comments/ questions:

  1. In Figure 1, the wall-coated immobilization in (1) and monolithic immobilization in (3) are clear and easy to understand from the current scheme. However, the packed bed in (2) can be confusing to readers – if by first sight regarding the blue spheres as only the enzymes. The current scheme may not well convey the existence of the packing beads and can be misunderstood as packing enzymes by themselves. The reviewer thinks it could be better illustrated by representing the enzymes and substrate separately rather than together as a blue sphere.
  2. In Figure 3, despite a, b and c being differentiated by LC and CE platforms, in c it further divides the detection into MS and UV. From the text, these different detections should also be available for LC methods. I would suggest only showing the LC and CE difference in the current figure without classifying detection and adding another scheme for different detections.
  3. Table 2 title is in a different font from the main text and captions

Reviewer 3 Report

See attachment for my feedback
